# Enhanced object detection in low-visibility haze conditions with YOLOv9s

**Yang Zhang, Bin Zhou** [ID]*, **Xue Zhao, Xiaomeng Song**

School of Computer Science and Technology, Shandong University of Technology, Zibo, Shandong, China

* freetzb@163.com

**Data availability statement:** The COCO dataset can be accessed at https://cocodataset.org, and both the RTTS and SOTS datasets, which are part of the RESIDE public dataset, can be

## Abstract

Low-visibility haze environments, marked by their inherent low contrast and high brightness, present a formidable challenge to the precision and robustness of conventional object detection algorithms. This paper introduces an enhanced object detection framework for YOLOv9s tailored for low-visibility haze conditions, capitalizing on the merits of contrastive learning for optimizing local feature details, as well as the benefits of multi-scale attention mechanisms and dynamic focusing mechanisms for achieving real-time global quality optimization. Specifically, the framework incorporates Patchwise Contrastive Learning to fortify the correlation among positive samples within image patches, effectively reducing negative sample interference and enhancing the model's capability to discern subtle local features of haze-impacted images. Additionally, the integration of Efficient Multi-Scale Attention and the Wise-IoU Dynamic Focusing Mechanism enhances the algorithm's sensitivity to channel, spatial orientation, and locational information. Furthermore, the implementation of a nonmonotonic strategy for dynamically adjusting the loss function weights significantly boosts the model's detection precision and training efficiency. Comprehensive experimental evaluations of the COCO2017 fog-augmented dataset indicate that the proposed algorithm surpasses current state-of-the-art techniques in various assessment metrics, including precision, recall, and mean average precision (mAP). Our source code is available at: https://github.com/PaTinLei/EOD.

## Introduction

With the intensification of industrialization and urbanization, air pollution and adverse weather conditions such as haze and sandstorms become increasingly common in urban settings. The phenomenon of extreme haze, in particular, poses significant risks to various sectors, including traffic management, urban security, and environmental monitoring. The precise identification of haze scenes is therefore crucial not only for enhancing the quality of urban life by safeguarding public safety and reducing traffic accidents, but also for contributing to the sustainable development of a green society.

The phenomenon of haze is prevalent in the atmosphere, and adversely affects imaging systems due to the scattering of light by atmospheric aerosols. This degradation in image quality can severely impact the accuracy of computer vision tasks, such as object detection. However,

accessed at https://sites.google.com/view/reside-dehaze-datasets/. These datasets are already hosted on stable, public repositories and are widely recognized in the research community.

**Funding:** This work was supported by the Natural Science Foundation of Shandong Province under Grant ZR2021MF031.

**Competing interests:** The authors have declared that no competing interests exist.

traditional object detection techniques, constrained by early detection technology, struggle to meet current demands. Although the YOLO deep neural network model has become a hot topic in the field of object detection due to its excellent real-time performance, accuracy, generalization capabilities, and global optimization quality, most existing networks are designed with clear or only moderately adverse weather conditions in mind. Their accuracy in object detection under extreme weather conditions, such as heavy haze, is still lacking.

To tackle this challenge, researchers have explored image dehazing enhancement prior to object detection. Eliminating haze from images not only improves image quality, but also enhances information readability, effectively boosting object detection effectiveness in low-visibility haze scenarios. Traditional image dehazing techniques, reliant on physical models and prior knowledge like Dark Channel Prior (DCP) [1], Color Attenuation Prior (CAP) [2], and Non-Local Color Prior (NCP) [3], have yielded results but suffer from limited model robustness and dehazing effects. Fortunately, image dehazing techniques based on deep learning have made significant progress, utilizing advanced methods like Convolutional Neural Networks (CNNs) [4], Generative Adversarial Networks (GANs [5]), Transformers [6], and Diffusion Models [7], offering the possibility of high-precision image quality restoration. However, end-to-end image dehazing models still have shortcomings in handling detail loss and color distortion, which restricts their direct application in object detection tasks, especially in low-visibility conditions.

To address the shortcomings of existing methods, this paper proposes a YOLOv9s [8] object detection framework tailored for low-visibility haze scenes. This framework integrates contrastive learning, dynamic focusing mechanisms, and attention mechanisms, aiming to maintain the real-time accuracy of the YOLOv9s algorithm while further enhancing its robustness in low-visibility haze environments. Specifically, the proposed method introduces Patchwise Contrastive Learning [9], which focuses on the correlation between positive samples within image patches, effectively suppressing the interference of negative samples. This strategy significantly improves the model's ability to capture local detail features in haze images, thus better preserving important image information during the dehazing process. Additionally, the proposed method introduces the Efficient Multi-Scale Attention [10] and Wise-IoU Dynamic Focusing Mechanism [11], which allocate attention weights at different scales, enabling the model to respond more sensitively to change in channel, spatial orientation, and positional information. This multi-scale feature fusion strategy enhances the model's understanding of complex scenes and improves the accuracy of object detection. To further enhance the model's detection accuracy and training efficiency, we employ the non-monotonic strategy for dynamically adjusting the weights of the loss function, and adaptively adjusting the weights of various terms in the loss function. This allows the model to focus on different optimization objectives at different training stages, further improving the model's detection accuracy and training efficiency. Extensive experimental results on the COCO2017 [12] dataset demonstrate that the detection algorithm proposed in this study outperforms existing state-of-the-art methods in multiple evaluation metrics, such as precision, recall, and mean average precision (mAP).

Specifically, the contributions of this paper are as follows:

- This paper proposes a YOLOv9s object detection framework tailored for low-visibility haze scenes, integrating contrastive learning, dynamic focusing mechanisms, and attention mechanisms. The proposed approach maintains the real-time and accuracy of the YOLOv9s algorithm while enhancing its robustness in low-visibility haze environments.
- Patchwise Contrastive Learning is introduced to focus on the correlation within image patches for positive samples, effectively suppressing the interference of negative samples,

significantly improving the model's ability to capture local detail features in haze images and better preserving crucial image information during the dehazing process.
- The Efficient Multi-Scale Attention and Wise-IoU Dynamic Focusing Mechanism are incorporated, allocating attention weights across different scales to render the model more responsive to variations in channel, spatial orientation, and positional information, enhancing the model's comprehension of complex scenes and improving the precision of object detection.
- A non-monotonic strategy is employed for dynamically adjusting the weights of the loss function, adaptively adjusting the weights of various terms in the loss function, allowing the model to focus on disparate optimization objectives at different training stages, further elevating the model's detection accuracy and training efficiency.

## Related work

### Image dehazing

Image dehazing technology is a method for processing images captured under hazy weather conditions. Its primary objective is to remove haze from images, thereby enhancing image quality and improving information readability. With the continuous advancements in computer vision (CV) and artificial intelligence, image dehazing technology has found widespread applications in image processing, autonomous driving, and other domains. Existing image dehazing techniques can be broadly classified into traditional methods and those based on deep learning.

### Traditional image dehazing technology

Traditional image dehazing technology involves analyzing the imaging mechanism and the degradation process under hazy weather conditions. This includes establishing corresponding physical models and performing inverse operations to restore clear images. The classic atmospheric scattering model proposed by McCartney [13] et al. serves as the foundation for image dehazing technology. Some scholars treat atmospheric light and scene depth [14] as prior conditions, obtain location information through interactive methods, and estimate model parameters accordingly. Narasimhan [15] et al. have proposed a multi-image dehazing algorithm that utilizes the radiance differences in different scenes and the polarization characteristics of light to achieve image restoration. Conversely, single-image dehazing methods can address the color deviation problem that may arise after applying multi-image algorithms. Oakley [16] et al. have achieved dehazing by using the relative transmittance of image pixel scattering and reflection fluxes. Furthermore, Iwamoto [17] et al. have addressed the time consumption issue in the original dark channel prior by optimizing pixel normalization, resulting in improved performance. He [18] et al. have proposed an image dehazing algorithm based on the dark channel prior (DCP) to overcome the limitations of physical models in achieving real-time processing, achieving notable results.

### Deep learning-based image dehazing technology

In recent years, the application of deep learning theory in image dehazing has made significant progress. Scholars have proposed various dehazing and quality evaluation algorithms utilizing convolutional neural networks (CNN), including DehazeNet, AOD-Net, MSCNN, and so on. Zhang [19] et al. propose HazDesNet, an innovative end-to-end convolutional neural network that predicts pixel-level haze density maps from hazy images. Min [20] et al. systematically evaluate and propose a novel quantitative quality evaluation method for dehazing

algorithms, leveraging a synthetic haze-removing quality (SHRQ) database and integrating specific dehazing features to accurately assess the performance of image dehazing algorithms, particularly tailored for aerial images. Min [21] et al. systematically study and propose an objective digital humanities quarterly index (DHQI) by integrating haze-removing, structure-preserving, and over-enhancement features to accurately evaluate the quality of image dehazing algorithms. DehazeNet, proposed by Cai [22] et al., uses multi-scale CNNs and max-pooling technology to estimate the transmission map and invert the haze-free image. However, the DehazeNet method has certain limitations, such as the inability to fuse deep and shallow information, insufficient feature extraction from hazy images, color distortion, and incomplete dehazing. AOD-Net, introduced by Li [23] et al., assumes that the atmospheric light value and light transmission rate are model parameters, reducing the accumulated errors caused by these parameters. MSCNN, proposed by Ren [24] et al., uses a coarse-scale network to estimate the scene transmission map, which is then refined by a fine-scale network. However, the MSCNN method performs poorly when dealing with images captured under night haze.

In addition to CNN-based methods, Generative Adversarial Networks (GAN) have also been introduced in image dehazing. Researchers have conducted extensive research on image dehazing methods based on GAN technology. For instance, Huang [25] et al. have proposed a stacked conditional generative adversarial network method that can remove haze from each RGB color channel and achieve fast convergence simultaneously. Chai [26] et al. have proposed a PDD-GAN dehazing network based on the PeleeNet architecture to address the issue of insufficient color recovery in CNNs when edge and prior visual guidance networks are absent. In summary, these GAN-based methods exhibit better performance in real image dehazing.

Deep learning-based image dehazing techniques incorporating attention mechanisms have also emerged in recent years. By mimicking the human visual system, these algorithms can focus on the key areas of an image, thereby improving the effectiveness of image dehazing. Ren [27] et al. have proposed a single-image dehazing method using a gate fusion network. They utilized an attention mechanism to extract image features in a CNN and then weighted those features to accurately restore the details obscured by haze. Liu [28] et al. have proposed a multi-scale guidance network utilizing the attention mechanism for single-image dehazing. By incorporating attention mechanisms, the algorithm can better focus on critical areas of the image, enhancing the accuracy of the dehazing method.

Furthermore, the application of an improved loss function can assist the network in better adapting to specific tasks and enhancing the quality of the recovered image. For example, Yin [29] et al. have introduced multi-perception loss and fidelity loss into GAN to maintain the network's focus on useful information in the image, reducing the impact of noise, and improving the quality of image recovery. Engelmann [30] et al. have employed Wasserstein distance and auxiliary classifier loss to enhance GAN, enabling the network to better handle class imbalance and improve the performance of generated samples in downstream classification tasks. Jiang [31] et al. have proposed a focal frequency loss, which complements spatial loss. This can effectively mitigate the loss of important frequency information that may be caused by the inherent bias of neural networks and also demonstrates potential advantages in StyleGAN2. The introduction of loss functions further enhances the stability of network training, the quality of generated samples, and the overall performance of the network.

In concluding our proposed methods for screen content quality assessment, it is equally important to recognize that other image processing tasks, such as image dehazing, also require effective quality assessment techniques. With the growing application of deep learning in image dehazing in recent years, accurately evaluating the quality of dehazed images has

become a crucial challenge. Min X and colleagues explore the differences between screen content and natural scenes [32,33], establishing benchmarks for screen and video quality assessment, and underscoring the challenges involved in evaluating screen content quality. Similarly, Zhai G and Min X conduct a survey on perceptual image quality assessment [33], categorizing various image quality metrics and comparing their performance, thus offering valuable insights into both traditional and contemporary methods for evaluating visual signal quality. By incorporating quality assessment methods from screen content, video, and visual signals, we can more comprehensively evaluate the effectiveness of image dehazing algorithms, particularly in terms of detail restoration and color accuracy.

## Contrastive learning

Contrastive learning has played a pivotal role in self-supervised and unsupervised representation learning domains [34]. In traditional self-supervised representation learning, contrastive learning is often applied to high-level vision tasks, inspired by the fundamental ideas of N-pair loss [35], noise-contrastive estimation [36], and triplet loss [37]. The most critical design aspect lies in the selection of positive and negative samples, as these tasks are inherently suited for modeling comparisons between positive and negative samples or features. The model optimizes the distribution of samples in the space by pulling positive samples closer to a given anchor point while pushing negative samples farther away. In traditional unsupervised learning, contrastive learning is commonly used in multi-sensory prediction, cross-modal encoding, and denoising tasks, with the primary goal of predicting another part of the data while preserving feature details. Specifically, this is achieved by learning a compressed encoding or representation [38] that reduces the amount of information while retaining enough crucial information to effectively reconstruct or restore the original input data when needed.

In recent years, contrastive learning has been applied to low-level vision tasks such as dehazing [39], deraining [40], image super-resolution [41], and image-to-image translation, achieving remarkable results. However, these methods have a limitation: they rely on a manually designed and defined loss function to evaluate the model's prediction performance, quantifying the accuracy or error of the model's predictions. This approach cannot fully reflect the model's performance across various scenarios.

To address this issue, contrastive learning methods attempt to introduce the maximization of mutual information [42]. In the representation space, they learn a data representation through noise-contrastive estimation techniques and gather related signals together, thereby separating these signals from other unrelated sample signals. These correlated signals may also be between an image and its original version, intra-neighboring regions, or downstream generated images. The introduction of this method can maximize the preservation of feature details in similar scenarios, particularly demonstrating superior performance in preserving details and image restoration in real-world hazy scenarios.

## Object detection

In recent years, YOLO, a popular object detection method particularly effective under hazy conditions, has experienced rapid development. YOLOv1 [43], published in 2016, integrates feature extraction, classification regression, and prediction into a single connected layer. Subsequently, YOLOv2 [44] was optimized with the introduction of a new backbone network, Darknet19, along with numerous new concepts and training strategies. Compared to its predecessors, YOLOv3 [45] further updated the backbone network to Darknet53 and incorporated FPN as the feature enhancement network, marking a milestone in the field of object

detection. YOLOv4 [46], proposed by Bochkovskiy et al., optimized the backbone network by adding a CSP module and introduced various strategies in the training stage, resulting in significantly improved detection rates on the Microsoft COCO2017 dataset [12]. YOLOv5 [47], released by Glenn Jocher in 2020, further refined YOLOv4 and achieved an average precision (AP) of 50.7% on the hazed COCO2017 dataset. In 2021, YOLOX [48] was published, featuring five major changes compared to YOLOv3: an anchor-free structure, multi-positivity, decoupled head, effective label assignment, and stronger augmentation. YOLOv6 [49], published in September 2022, offers various model sizes suitable for industrial applications, similar to YOLOv4 and YOLOv5. Based on VGGnet with re-parameterization (RepVgg), YOLOv6 is more suited for higher parallelism than previous YOLO versions. YOLOv7 [50], also released in 2022 by Bochkovskiy et al., surpasses popular object detectors in terms of speed and accuracy. It adopts trainable bag-of-freebies for real-time detection with higher accuracy, extended efficient layer aggregation networks (E-ELAN), model scaling based on concatenation, and introduces planned re-parameterization convolution and other techniques. These improvements enhance YOLOv7's accuracy without compromising its estimating speed. By 2023, the latest version, YOLOv8 [51], was released by Ultralytics. The YOLOv8 model has an updated architecture and adopts anchor-free detection technology, demonstrating excellent performance in real-time detection with both speed and accuracy. YOLOv9 [8]is the latest iteration in the YOLO (You Only Look Once) family of object detection models, released in February 2024. It introduces several innovations aimed at improving both efficiency and accuracy. YOLOv10 [52], released in May 2024, builds on the advancements of YOLOv9 with further improvements in speed. However, it lacks sufficient stability. In 2024, Y. Azadvatan et al. proposed a real-time deep learning algorithm for object detection, the Mel-Net method [53], which further improves the real-time performance of single-stage object detection models.

Beyond the YOLO series, other object detection methods have continued to evolve and innovate. One of the most representative approaches is the family of region-based convolutional neural networks (R-CNN). From 2010 to 2012, the performance of traditional object detection methods, which relied on manually extracted features, had reached a plateau. In 2014, R. Girshick et al. [54]broke this stagnation by introducing the Regions with CNN features (R-CNN) model, marking a pivotal turning point that sparked rapid advancements in object detection. In 2015, R. Girshick further proposed Fast R-CNN [55], which improved speed by over 200 times compared to R-CNN, though it was still constrained by the region proposal bottleneck. In 2020, Zhu et al. introduced Deformable DETR [56], a novel Transformer-based architecture that demonstrated robust performance in complex scenarios. In 2021, T. Panboonyuen et al. improved the performance of the YOLO series models for object detection of road assets on Thailand highway panoramas by using a Transformer-based YOLOX and a feature pyramid decoder [57]. By 2023, Chen et al. built upon the progress in this field by proposing DiffusionDet [58], a framework based on diffusion models, which further elevated detection accuracy to new heights. However, despite these advancements, these methods face limitations in real-time applications due to their computational complexity and latency issues.

In object detection, significant attention has been garnered by the methods that incorporate attention mechanisms into YOLO. By introducing attention mechanisms, YOLO's accuracy and speed can be further enhanced, augmenting its perception and localization capabilities for critical targets. For instance, the ECA-Res2Net-YOLO model in VGG-style [59] replaces the original DBL structure with a layered residual structure, Res2Net, to bolster multi-scale feature extraction. It also employs the highly efficient channel attention mechanism, ECA, to capture dependency relationships between adjacent channel features, thereby

improving the network's attention to pedestrian targets. Guo et al [60]. have fused a CA attention mechanism based on YOLOv5, resulting in a weed model with heightened classification and detection capabilities, effectively addressing the high similarity between weeds and crops. These methodologies improve detection performance and accuracy by intensifying the network's focus on critical targets and key features.

Furthermore, refining the loss function offers a novel direction for enhancing YOLO's accuracy. Wang et al. [61] have proposed a new boundary box similarity measure, substituting the standard IoU with Wasserstein distance. This provides a more effective and adaptable measurement method for detecting small-scale objects and demonstrates good performance. Hu et al. [62] have incorporated focal loss into YOLOv4 to tackle the category imbalance problem, achieving notable improvements. Wang et al. [63] have replaced the original bounding box loss function in YOLOv7 with SIoU, further enhancing detection accuracy by redefining the penalty term. Huang et al. [64] have proposed an improved DC-SPP-YOLO model based on YOLOv2, introducing an updated loss function combining mean square error (MSE) and cross-entropy loss, which has yielded favorable results.

## The proposed method

Our approach introduces an enhanced object detection framework tailored for YOLOv9s, specifically designed to address low-visibility haze conditions. The proposed method primarily consists of two components: image preprocessing and object detection. In the image preprocessing stage, this paper employs a GAN-based image dehazing algorithm [65], on this basis, further refines by incorporating Patchwise Contrastive Learning. This integration fortifies the correlation among positive samples within image patches, effectively mitigating negative sample interference and enhancing the model's ability to discern subtle local features in haze-impacted images, ultimately yielding clearer and dehazed images. Following the generation of these clearer images, YOLOv9s is applied for object detection, with further enhancements introduced through the integration of Efficient Multi-Scale Attention and the Wise-IoU Dynamic Focusing Mechanism. This integration enhances the algorithm's sensitivity to channel, spatial orientation, and locational information. Additionally, the implementation of a non-monotonic strategy for dynamically adjusting loss function weights significantly boosts the model's detection precision and training efficiency. Our main framework diagram is shown in Fig 1.

### Dehazing model

In this study, an improved GAN based on patchwise contrastive loss was used to enhance image features, effectively improving the quality and accuracy during the image restoration process. The model achieves efficient feature transformation from input images to output images by integrating an encoder and multiple perceptrons. Initially, the encoder transforms the input images into a high-dimensional feature space, a crucial step as it lays the foundation for subsequent feature processing. During feature extraction, the multi-layer perceptrons further analyze and refine these features to capture the details and complexity within the images. This deep feature learning mechanism is implemented through multiple hidden layers, each enhancing the features extracted by the previous layer. Additionally, the model incorporates a patchwise contrastive loss, which significantly improves the clarity and visual quality of the images by comparing the feature differences in local areas between the input and output images. This type of contrastive loss not only strengthens the model's ability to capture image details but also optimizes the overall consistency of the generated images, making the restored

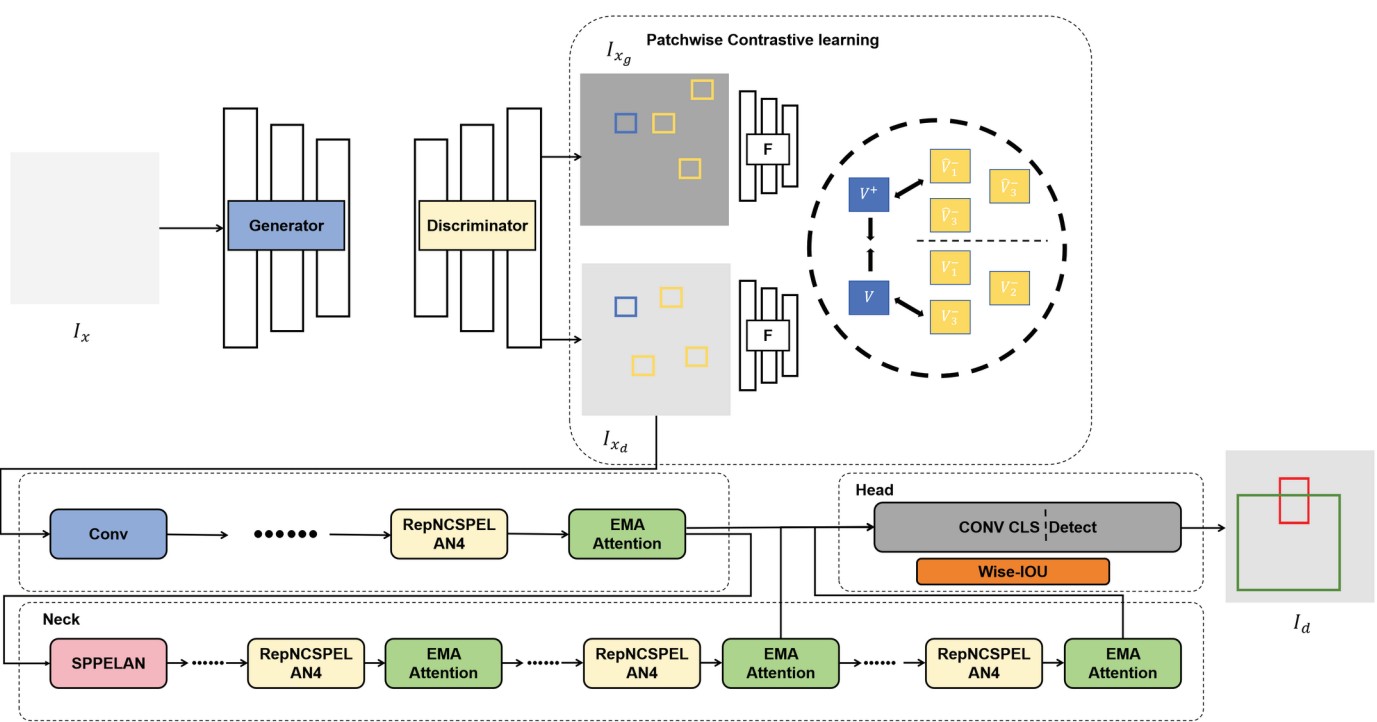

**Fig 1. The network framework of the proposed model.** $I_x$ represents the foggy image, $I_{x_d}$ is the dehazed image of $I_x$, $I_{x_g}$ is the clear image corresponding to $I_x$, and $I_d$ is the final object detection result.

images appear more natural and realistic visually. The Patchwise Contrastive Learning [9] is illustrated in Fig 2.

The idea of contrastive learning is to associate the information of two pictures, which can better learn the mutual information. We map the query, positive number, and N negative numbers to a k-dimensional vector, i.e. $v$, $v^+ \in \mathbb{R}^K$, and $v_n^- \in \mathbb{R}^K$ is the $n^{th}$ negative number. We formulate a $(N + 1)$ classification problem and compute the probability that a "positive" is chosen over "negatives". The definition of the PatchNCE loss is written as Eq 1.

$$\mathcal{L}_{\text{PatchNCE}} = -\log\left(\frac{\exp\left(\text{sim}\left(v, v^+\right)/\tau\right)}{\exp\left(\text{sim}\left(v, v^+\right)/\tau\right) + \sum_{n=1}^{N}\exp\left(\text{sim}\left(v, v_n^-\right)/\tau\right)}\right) \qquad (1)$$

where $\text{sim}(a, b)$ denotes the similarity function between vectors $a$ and $b$, commonly defined as the cosine similarity. $\tau$ is a temperature parameter that controls the sharpness of the distribution.

## Object detection

After image dehazing, YOLOv9s is applied for object detection, with further enhancements achieved through the integration of Efficient Multi-Scale Attention [10] and the Wise-IoU Dynamic Focusing Mechanism [11]. Initially, we introduce these two mechanisms: Efficient Multi-Scale Attention and Wise-IoU Dynamic Focusing Mechanism. This integration augments the algorithm's sensitivity to channel, spatial orientation, and locational information. Furthermore, the implementation of a non-monotonic strategy for dynamically adjusting

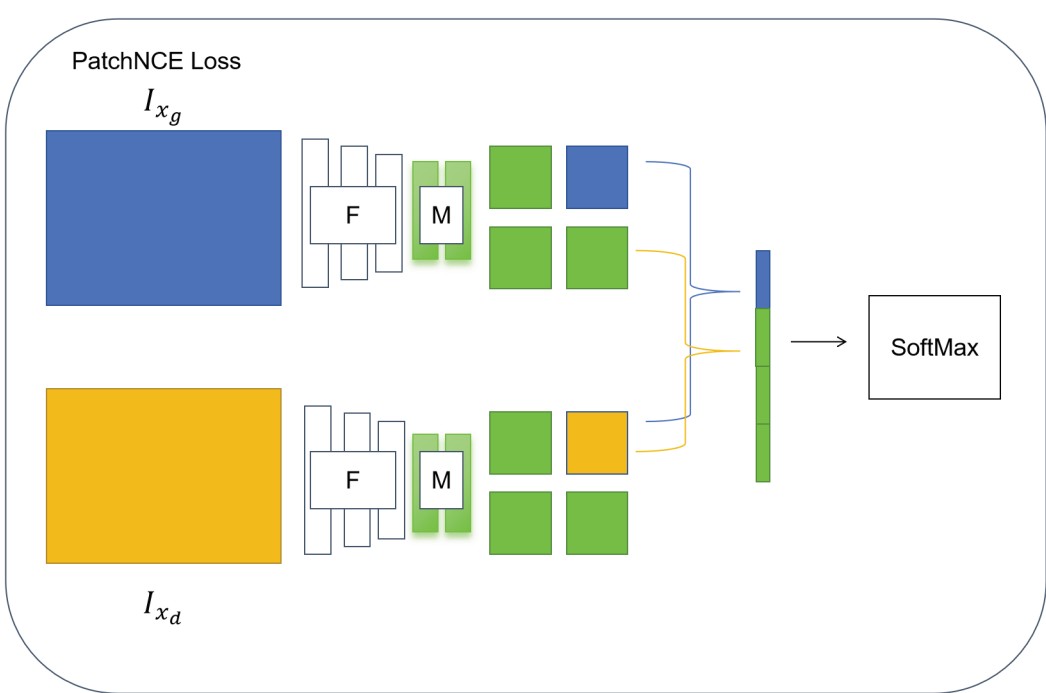

**Fig 2. The patchwise contrastive learning method based on PatchNCE loss.**

loss function weights significantly enhances the model's detection precision and training efficiency. The diagram of the improved YOLOv9s model is presented in Fig 3.

**Efficient multi-scale attention.** Attention mechanisms are widely applied in deep learning, with common types including spatial, channel, and hybrid attention mechanisms. These mechanisms mimic human visual and cognitive systems by selectively concentrating on key information in the input, thereby enhancing the model's performance and generalization capabilities.

The Efficient Multi-Scale Attention (EMA) module is an effective multi-scale attention mechanism designed to perform cross-spatial learning without reducing dimensions. It leverages a grouped structure and cross-spatial learning methods to establish short-term and long-term dependencies through a design of multi-scale parallel sub-networks, thus improving the detector's performance while reducing parameter requirements and computational costs. The process involves dividing the input feature map along the channel dimension into $G$ sub-feature maps, integrating three parallel sub-networks. Specifically, the first two parallel sub-networks are located in the 1×1 branch, while the third is in the 3×3 branch. The first two parallel sub-networks, similar to the Channel Attention (CA) mechanism, involve two one-dimensional average pooling operations, concatenation, and a 1×1 convolution to facilitate the learning of diverse semantic features. By integrating context information from 1×1 and 3×3 scales, pixel-level attention on high-level feature maps is enhanced. Cross-spatial learning produces two output feature maps, and the spatial attention weight sum within each group is calculated. Ultimately, this sum undergoes a Sigmoid operation, and the final output feature maps are obtained through multiplication. The EMA module significantly enhances the detection performance of YOLOv9s, especially in complex backgrounds. The Efficient Multi-Scale Attention in Fig 4.

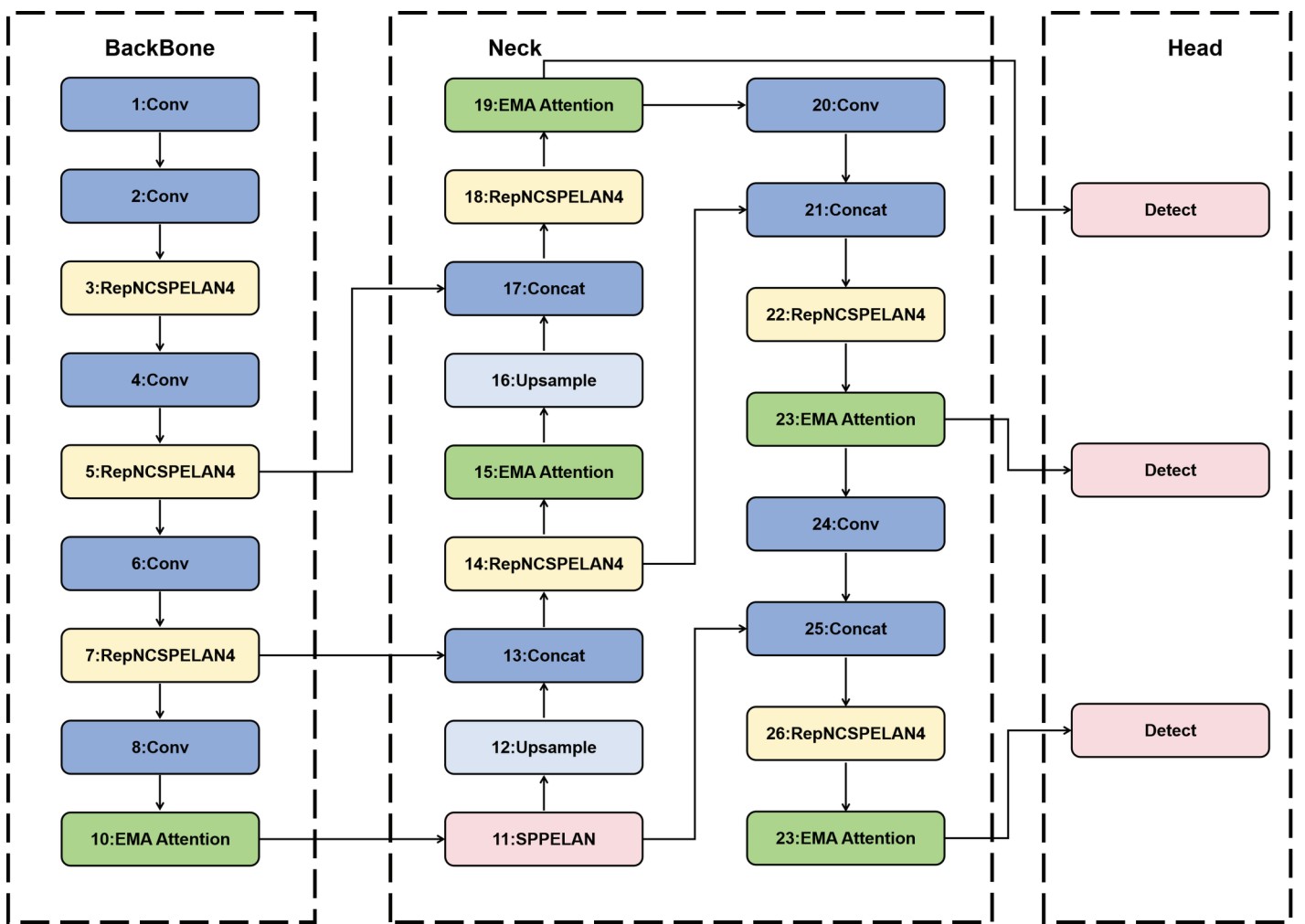

**Fig 3. The improved network diagram of YOLOv9s.**

The loss function of the improved YOLOv9s includes the BCE loss, the distribution focal loss, and the Wise-IoU loss.

**BCE loss.** BCE Loss, or Binary Cross-Entropy Loss, helps address class imbalance. In object detection tasks, the background (negative samples) often significantly outnumbers the objects (positive samples). By balancing the loss weights between positive and negative samples, BCE Loss ensures that the loss from negative samples does not dominate the training process. The focal loss is used to address the issue of focus loss in BCE Loss. The key features of BCE Loss are as follows: BCE Loss employs the Sigmoid function to convert logits into probability values, representing the probability of positive samples. BCE Loss is a loss function used for binary classification problems to evaluate the consistency between the model's output and the actual labels, while also having the ability to handle class imbalance. The parameter $p_{cls}$ represents the predicted class probability, which ranges between 0 and 1. The parameter $t_{cls}$ denotes the true class label, typically being 0 or 1. The $L_{BCE}$ function is shown in Eq 2.

$$L_{BCE}(p_{cls}, t_{cls}) = -\left(t_{cls} \cdot \log(p_{cls}) + (1 - t_{cls}) \cdot \log(1 - p_{cls})\right) \tag{2}$$

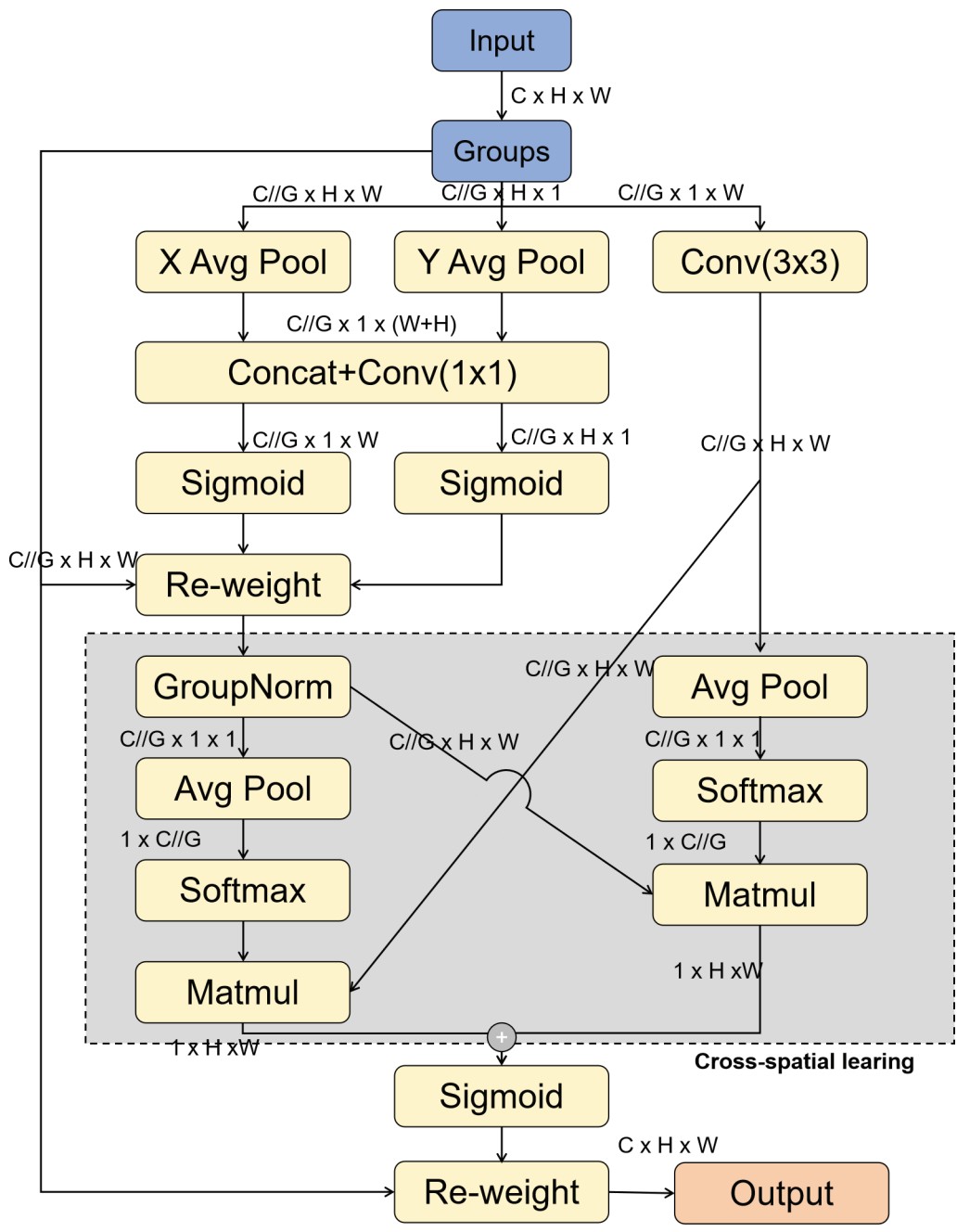

**Fig 4. The efficient multi-scale attention.**

**Distribution focal loss.** Modeling the position of a bounding box as a general distribution is a strategic innovation in object detection. By representing the bounding box position as a distribution rather than a fixed point, it empowers the network with the ability to adapt swiftly to diverse scenarios. This dynamic approach enables the network to prioritize the region of interest, concentrating on the location distribution that closely aligns with the actual target position distribution. This adaptability is particularly valuable in object detection tasks

where the target's location may vary within an image due to factors like scale, orientation, or occlusion. As a result, the model becomes more robust and efficient, making it well-equipped to handle complex and dynamic scenes, ultimately increasing the accuracy of object detection. The parameter $p_{\text{dfl}}$ represents the predicted bounding box location distribution, while $t_{\text{dfl}}$ is the true bounding box location distribution. Specifically, $p_{\text{dfl},j,\text{left}}$ and $t_{\text{dfl},j,\text{left}}$ are the predicted and true left boundary distribution values for the $j^{\text{th}}$ bounding box, respectively. Similarly, $p_{\text{dfl},j,\text{right}}$ and $t_{\text{dfl},j,\text{right}}$ are the predicted and true right boundary distribution values for the $j^{\text{th}}$ bounding box, respectively. The $L_{\text{DFL}}$ function is shown in Eq 3.

$$L_{\text{DFL}}(p_{\text{dfl}}, t_{\text{dfl}}) = \sum_{j=1}^{n} \left( \text{H}(p_{\text{dfl},j,\text{left}}, t_{\text{dfl},j,\text{left}}) + \text{H}(p_{\text{dfl},j,\text{right}}, t_{\text{dfl},j,\text{right}}) \right) \tag{3}$$

where H represents the cross entropy loss function.

**Wise-IoU loss.** Most research assumes that the training data samples are of high-quality, and therefore focuses on improving the fitness of the boundary box. However, we have noticed that object detection training data may contain low-quality samples. Reinforcing the boundary box regression for low-quality samples will obviously damage the performance of the detection model. To solve this, Focal-EIoU is proposed. However, its focusing mechanism is static and fails to make full use of the non-monotonic focusing mechanisms. Therefore, we propose a dynamic nonmonotonic focusing mechanism that uses the Wise-IoU (WIoU) loss. The dynamic non-monotonic focusing mechanism uses outliers to measure the quality of anchor boxes instead of using IoU and combines a smart gradient gain allocation strategy. Therefore, it decreases the importance of high-quality anchor boxes while bringing down harmful gradients generated by low-quality samples. This enables WIoU to pay attention to the anchor boxes of low quality, thus increasing the accuracy of the detector. The schematic diagram of WIoU is in Fig 5.

IoU is adopted to evaluate the degree of overlap between the predicted bounding boxes and the ground truth boxes in object detection. It is calculated as Eq 4.

$$\mathcal{L}_{\text{IoU}} = 1 - \text{IoU} = 1 - \frac{W_i H_i}{S_u} \tag{4}$$

where $W_i$ and $H_i$ are the dimensions of the overlapping regions. $S_u$ is the joint area, $S_u = WH + W_{gt}H_{gt} - W_i H_i$, $W$ and $H$ are the dimensions of the anchor box, $W_{gt}$ and $H_{gt}$ are the dimensions of the target box.

We design this dynamic non-monotonic focus mechanism by IoU and measure the quality of the anchor box by the outlier value. The outlier value $\beta$ is defined as Eq 5:

$$\beta = \frac{\mathcal{L}_{\text{IoU}}^*}{\mathcal{L}_{\text{IoU}}} \in [0, +\infty] \tag{5}$$

where $\mathcal{L}_{\text{IoU}}^*$ is monotonic focus mechanism.

The hyperparameters $\alpha, \delta$ for creating a non-monotonic focusing coefficient for WIoU are as follows in Eq 6:

$$\mathcal{L}_{WIoU} = r \left( \exp \left( \frac{(x - x_{gt})^2 + (y - y_{gt})^2}{W_g^2 + H_g^2} \right) \left( 1 - \frac{W_i H_i}{S_u} \right) \right),$$
$$r = \frac{\beta}{\delta \alpha^{\beta - \delta}} \tag{6}$$

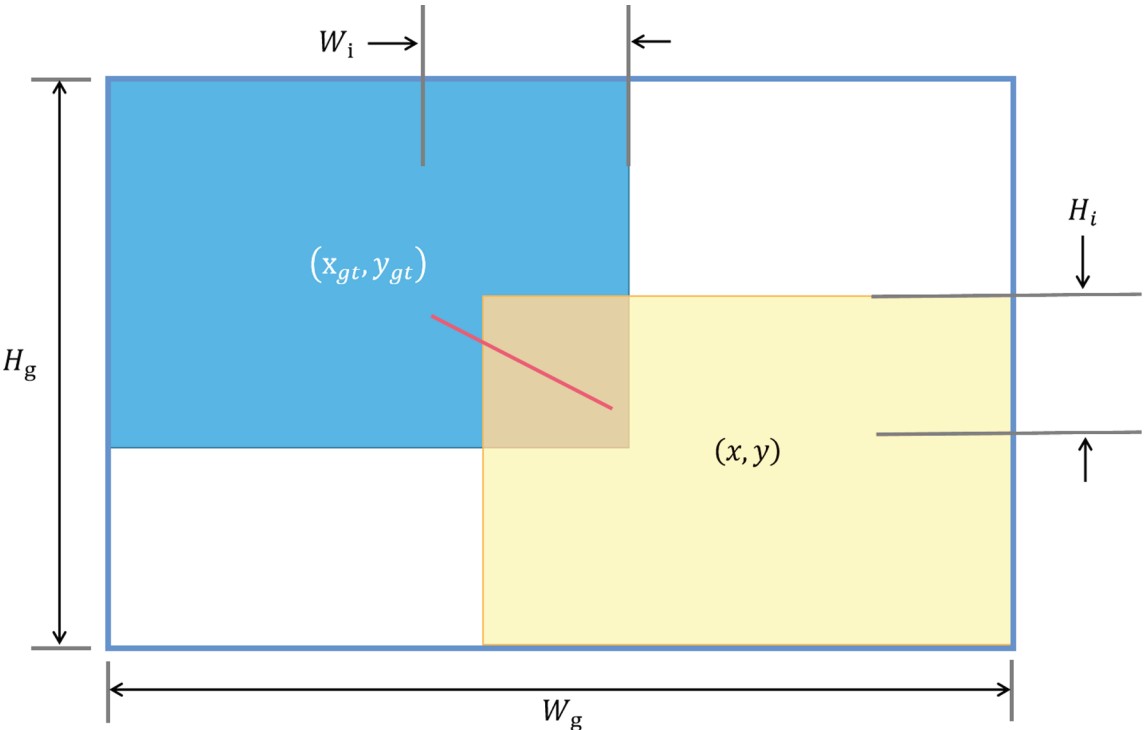

**Fig 5. IoU measures the overlap between predicted bounding boxes and ground truth boxes.** It is defined as the intersection area divided by the union area of the two boxes. WIoU implements a dynamic, non-monotonic focusing mechanism by IoU.

where $W_g$, $H_g$ are the size of the smallest bounding box, $x$, $y$ are the center coordinates of the anchor box, and $x_{gt}$, $y_{gt}$ are the center coordinates of the target box.

The total loss function of the improved YOLOv9s is shown in Eq 7.

$$
\begin{aligned}
\mathcal{L} = {} & \lambda_{\text{box}} \cdot \frac{1}{N} \sum_{i=1}^{N} \left( 1 - \mathcal{L}_{WIoU}(B_i, \hat{B}_i) \right) \\
& + \lambda_{\text{cls}} \cdot \frac{1}{N} \sum_{i=1}^{N} \mathcal{L}_{BCE}(p_{\text{cls}}^i, t_{\text{cls}}^i) \\
& + \lambda_{\text{dfl}} \cdot \frac{1}{N} \sum_{i=1}^{N} \mathcal{L}_{DFL}(p_{\text{dfl}}^i, t_{\text{dfl}}^i),
\end{aligned}
\tag{7}
$$

where $\lambda_{\text{box}}$ is the weight hyperparameter for the box loss, $\lambda_{\text{cls}}$ is the weight hyperparameter for the classification loss, $\lambda_{\text{dfl}}$ is the weight hyperparameter for the distribution focal loss, $N$ is the number of samples, WIoU$(B_i, \hat{B}_i)$ is the Wise Intersection over Union between the $i^{\text{th}}$ predicted box $B_i$ and the ground truth box $\hat{B}_i$, where $B_i = [x, y, w, h]$, $\hat{B}_i = [x_{gt}, y_{gt}, w_{gt}, h_{gt}]$. BCE$(p_{\text{cls}}^i, t_{\text{cls}}^i)$ is the binary cross-entropy loss between the $i^{\text{th}}$ predicted class scores $p_{\text{cls}}^i$ and the ground truth class labels $t_{\text{cls}}^i$, and DFL$(p_{\text{dfl}}^i, t_{\text{dfl}}^i)$ is the distribution focal loss between the $i^{\text{th}}$ predicted distribution $p_{\text{dfl}}^i$ and the ground truth distribution $t_{\text{dfl}}^i$.

## Experiment

### Network implementation

In our experiment, we use the improved YOLOv9s as the object detector for the purpose of fair and comprehensive evaluation. During the training process, data augmentation techniques like random flip or scale, and HSV augmentation are applied. The initial batch size is 16, and the learning rate is 0.0001. The training is performed for 300 epochs using the SGD optimizer with a cosine learning rate, and an early stopping mechanism is employed to prevent overfitting. Our haze detection method is run in the Pytorch framework on the NVIDIA GeForce RTX 3080 GPU.

### Datasets

The Microsoft COCO2017 dataset is a publicly available and widely used comprehensive dataset for image processing and computer vision research, containing 118,000 training images and 5,000 test images. This dataset is highly regarded for its diversity and complexity, featuring a wide range of objects in various contexts, making it particularly suitable for tasks such as object detection, image segmentation, and image captioning. The COCO2017 dataset includes numerous object categories, with each image annotated with a large number of instances, significantly enhancing its value for robust model training. The dataset can be accessed via the following link:https://cocodataset.org.

In the context of image dehazing, which involves removing haze or fog from images to improve clarity, the Microsoft COCO2017 dataset provides a rich base for training. In this paper, we apply the haze preprocessing method based on single view depth prediction [66] to the COCO2017 data set. This method uses an improved Multi-View Stereo (MVS) algorithm based on Collision-Mapping (COLMAP) to ensure geometric consistency between adjacent depth maps at each stage by iteratively calculating depth maps, eliminating the effect that background depth may "erode" foreground objects, and obtaining accurate depth images. Then, by adjusting the transmittance and atmospheric parameters and considering the scattering, absorption, and propagation of light, the scattering effect of the real foggy scene can be simulated by estimating the fog concentration of each pixel according to the information in the depth image, thus the generated foggy image is closer to the real foggy scene.

In addition, to further verify the completeness and generalization ability of the experiment, the Realistic Single Image Dehazing (RESIDE) dataset is introduced. This dataset, widely used for image dehazing research, is created by researchers at Fudan University in China specifically for developing and evaluating dehazing algorithms. RESIDE simulates real-world hazy conditions and includes a large number of synthetic and real foggy images. In subsequent experiments, two publicly available subsets of the RESIDE dataset, the Real Task-driven Testing Set (RTTS) and the Synthetic Object Testing Set (SOTS), are used to supplement the evaluation of object detection methods and image dehazing techniques. The RESIDE dataset can be accessed via the following link: https://sites.google. com/view/reside-dehaze-datasets/. The links for RTTS and SOTS are as follows: https:// utexas.app.box.com/s/2yekra41udg9rgyzi3ysi513cps621qz https://utexas.app.box.com/s/ uqvnbfo68kns1210z5k5j17cvazavcd1.

### Quantitative evaluation

In object detection, several evaluation metrics are crucial for assessing the performance of models. Precision (P) is a metric that measures the proportion of correctly identifying positive cases among all cases that the model is classified as positive. It's crucial in scenarios where

the cost of false positives is high. Recall (R), on the other hand, quantifies the proportion of actual positives identified correctly by the model. It's particularly important in situations where missing a positive case could have significant consequences.

The mean Average Precision (mAP) is another critical metric in object detection. It combines precision and recall into a single value, providing an overall measure of the model's accuracy across different thresholds. This metric is especially useful for comparing the performance of different models on the same dataset. The mAP metric is typically specified at different threshold levels, such as mAP50 and mAP75. mAP50 refers to the mean Average Precision calculated at an Intersection over the Union (IoU) threshold of **50%**, and it is widely used to assess a model's general object localization capability. mAP75, on the other hand, demands that the model maintains high detection accuracy at a more stringent IoU threshold of **75%**. Consequently, mAP75 is a more rigorous metric, suited for tasks where the precise localization of objects.

In image dehazing tasks, two common evaluation metrics are Peak Signal-to-Noise Ratio (PSNR) and Structure Similarity Index Measure (SSIM). PSNR measures the difference between the dehazed image and the original image, with higher PSNR values indicating better dehazing performance and greater retention of image details. SSIM, in contrast, focuses on the structural fidelity of the image. It not only considers pixel-level differences but also takes into account luminance, contrast, and structural information. A higher SSIM value indicates that the dehazing algorithm better preserves the overall structure and perceptual quality of the image.

## Experimental results

Our primary objective is to excel in target detection under challenging haze conditions. We have carefully selected the hazed COCO (Common Objects in Context) dataset as the foundation for our experiments. However, it's essential to note that the hazed COCO2017 dataset itself presents unique challenges due to its atomized nature, which means that objects within images are often partially obscured, making target detection in hazy conditions even more demanding. On the other hand, the hazed COCO2017 dataset itself has a rich set of scenarios and numerous detection categories that can fulfill our experiments.

To verify our network, we have undertaken a series of experiments, differentiating between the atomized dataset and the original dataset. This allows us to evaluate our dehazing model comprehensively. The experimental results demonstrate that our approach achieved the best results across all YOLO and non-YOLO series. Moreover, through ablation studies, we have proven that the introduction of Patchwise Contrastive Learning, Efficient Multi-Scale Attention, and Wise-IoU Dynamic Focusing effectively improves YOLOv9s. We conduct multiple validations of the model and ultimately select the average of ten results, keeping the standard deviation within **0.5%**. Compared to the original version, our method has significantly increased P, R, and mAP by **4.4%**, **3.1%**, and **4.1%**, respectively.

In summary, our research endeavors, grounded in the hazed COCO2017 dataset, highlight the success of our dehazing model in enhancing target detection accuracy, thereby contributing to the advancement of computer vision technologies in challenging environmental contexts. The summarized results are presented in Table 1. From the visual outcomes of target detection, our method achieves the best performance across all YOLO series, regardless of the target size. Compared to non-YOLO series, our method achieves the best performance in small target detection.

**Table 1. Results on the hazed COCO2017 dataset.**

| Method | P | R | mAP | mAP50 | mAP75 |
|---|---|---|---|---|---|
| YOLOvXs [48] | – | – | 31.1 | 46.8 | 35.0 |
| YOLOv5s [47] | 57.2 | 41.0 | 28.7 | 44.3 | 31.7 |
| YOLOv8s [51] | 59.2 | 44.1 | 32.6 | 47.5 | 35.3 |
| YOLOv10s [52] | 60.1 | 45.3 | 33.2 | 49.0 | 37.0 |
| DiffusionDet(res50) [58] | – | – | 22.9 | 42.4 | 26.1 |
| DiffusionDet(res101) [58] | – | – | 27.3 | 43.2 | 30.8 |
| DETR(res101) [56] | – | – | 23.3 | 40.2 | 26.4 |
| DINO [67] | – | – | 35.1 | 51.5 | 39.2 |
| YOLOv9s [8] | 59.6 | 45.0 | 33.1 | 48.6 | 36.3 |
| Ours | 64.0 | 48.1 | 36.9 | 52.7 | 40.5 |

To validate the detection accuracy of the model in real-world low-visibility environments, we conduct experiments on the RTTS dataset, which the results present in Table 2. The findings indicate that our method achieves a **3.1%** improvement in mAP50 over the original YOLOv9s model and a **3.3%** improvement over the latest YOLOv10s model.

However, our method is also subject to low-visibility environments during object detection, which can be visualized through a confusion matrix. The object detection confusion matrix based on the hazed COCO2017 dataset is shown in Fig 6. In the confusion matrix, each row of the matrix represents a real category, and each column of the matrix represents a predicted category. Each row or column of the matrix contains a number of color blocks, each color block represents a certain proportion, the proportion value corresponds to the right-most color bar, the darker the color, the higher the percentage. The color of each color block is roughly divided into two categories: blue and red. In each row of the matrix, the blue color block represents the percentage of the number of samples predicted correctly to the actual total number of samples in each category, and the red color block represents the percentage of the number of samples predicted incorrectly or missed to the actual total number of samples in each category. In each column of the matrix, the blue color block represents the percentage of the number of samples predicted correctly to the total number of predicted samples in each category, and the red color block represents the percentage of the number of samples predicted incorrectly or missed to the total number of predicted samples in each category. From this figure, it is evident that the correct detection counts for large objects like elephant, bears, zebra, and giraffes are significantly higher than for smaller objects like cups, forks, knives, and spoons, and the former also has notably fewer missed detections than the latter.

**Table 2. Results on the hazed RTTS dataset.**

| Method | P | R | mAP50 |
|---|---|---|---|
| YOLOvXs | – | – | 60.2 |
| YOLOv5s | 71.9 | 54.3 | 58.8 |
| YOLOv8s | 73.7 | 55.9 | 59.5 |
| YOLOv10s | 75.0 | 58.2 | 61.9 |
| DiffusionDet(res50) | – | – | 55.4 |
| DiffusionDet(res101) | – | – | 59.3 |
| DETR(res101) | – | – | 55.7 |
| DINO | – | – | 63.6 |
| YOLOv9s | 74.6 | 59.6 | 62.1 |
| Ours | 78.4 | 61.3 | 65.2 |

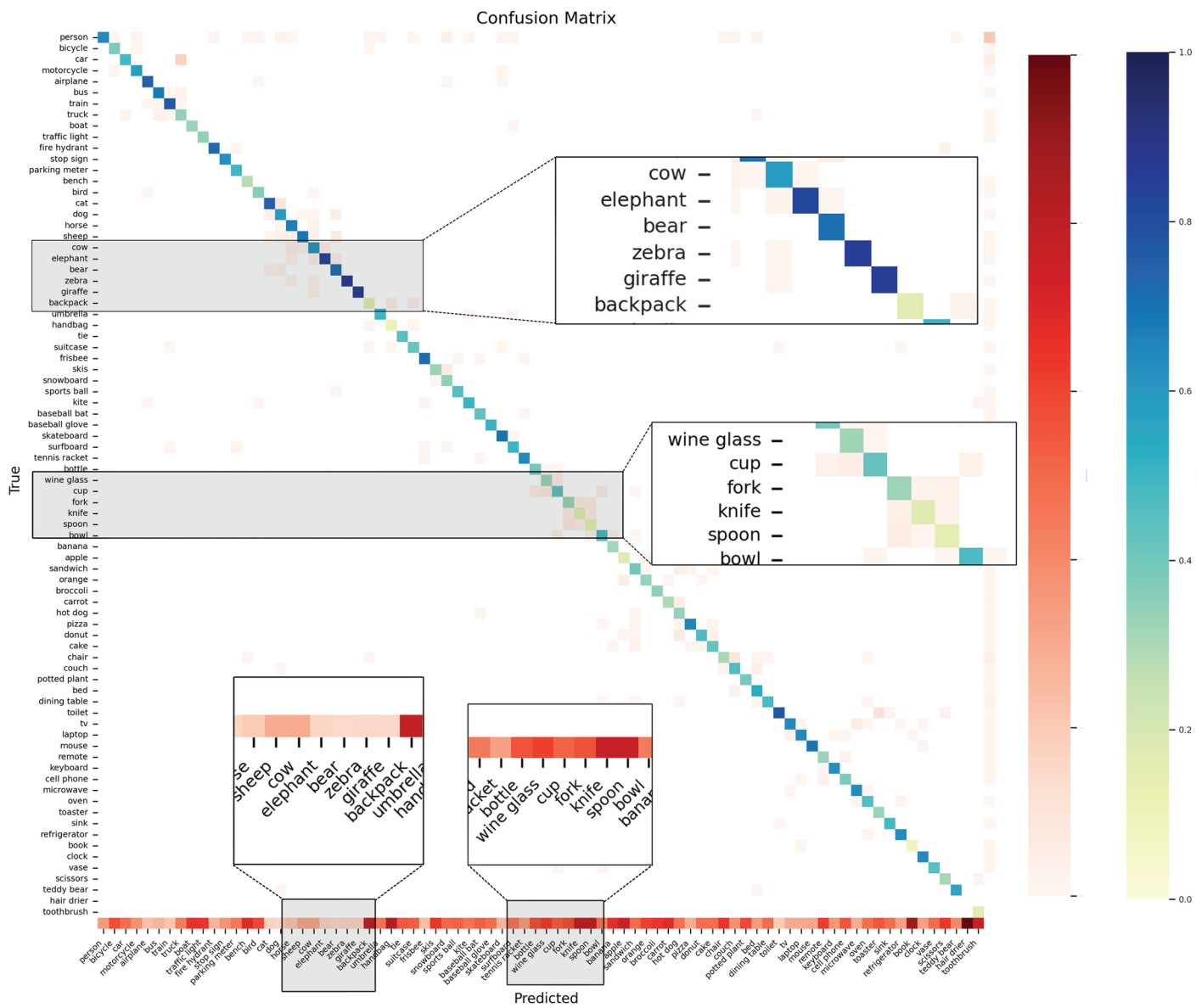

**Fig 6. The confusion matrix of our model in the hazed COCO2017 dataset.**

In addition, we establish various metrics to evaluate the performance of our model in comparison to other models. Performance testing is conducted using different object detection models combined with the dehazing model. A summary of the results is presented in Table 3. The findings indicate that our model is slightly slower than the original YOLOv9 in terms of FLOPs and training time, although the differences are minimal. Overall, while our model lags slightly behind other YOLO series in FLOPs, training time, and model size, it outperforms the non-YOLO series.

To validate the effectiveness of our dehazing module, we conduct comparative experiments on outdoor scenes from the SOTS public haze dataset. The findings indicate that our method outperforms traditional deep learning-based dehazing models such as GAN, with a 7-point

**Table 3. Model performance.**

| Method | FLOPS(ms) | Training Time(h) | Size(MB) |
|---|---|---|---|
| YOLOvXs | 139.0 | 6.0 | 106.8 |
| YOLOv5s | 74.2 | 3.5 | 53.0 |
| YOLOv8s | 108.5 | 5.0 | 59.6 |
| YOLOv10s | 95.4 | 4.0 | 69.5 |
| DiffusionDet(res50) | 343.2 | 22.0 | 533.7 |
| DiffusionDet(res101) | 488.9 | 27.5 | 461.0 |
| DETR(res101) | 427.9 | 25.0 | 275.0 |
| DINO | 550.0 | 30.0 | 548.6 |
| YOLOv9s | 142.7 | 12.0 | 115.1 |
| Ours | 152.1 | 14.5 | 115.1 |

increase in PSNR and a **13.9%** improvement in SSIM. Compared to the latest diffusion models, our method achieves a 1.1-point increase in PSNR and a **3.7%** improvement in SSIM. Our method outperforms other dehazing techniques, with detailed results presented in Table 4.

## Ablation experiment

**Influence of patchwise contrastive learning.** The introduction of Patchwise Contrastive Learning in the model was a pivotal enhancement, designed to concentrate on the relationships within local image patches for positive samples. This strategic approach effectively mitigated the disruptive impact of negative samples. As illustrated in Table 5, the incorporation of this learning method yielded a noticeable improvement in the model's performance on the hazed COCO2017 dataset. Specifically, the precision (P) metric saw a **1.4%** increase, the recall (R) metric improved by **0.8%**, and the mean average precision at mAP50 experienced a **1.1%** enhancement. These gains underscore the significant contributions of Patchwise Contrastive Learning to the model's ability to accurately detect objects in challenging, low-visibility conditions. In addition, when compared with the combination of the diffusion image dehazing method and YOLOv10, our method still achieved impressive results, with the precision (P) metric improving by **4.0%**, the recall (R) metric increasing by **3.8%**, and the mean Average Precision at mAP50 rising by **4.2%**.

The ablation study revealed that by incorporating PatchLoss, the model's ability to capture local detail features in haze images was significantly improved, and then the model can perform more precise object localization and classification, thereby improving the overall outcome of the target detection task.

**Table 4. Image dehazing.**

| Method | PSNR | SSIM |
|---|---|---|
| DCP [18] | 14.837 | 0.762 |
| CAP [2] | 18.194 | 0.781 |
| AODNet [23] | 19.415 | 0.804 |
| CycleGAN [68] | 11.107 | 0.649 |
| GAN [5] | 23.207 | 0.823 |
| SDAGAN [65] | 24.941 | 0.861 |
| Transformer [6] | 28.194 | 0.884 |
| Diffusion [7] | 29.081 | 0.925 |
| Ours | 30.217 | 0.962 |

**Table 5. Influence of PatchLoss.**

| Method | P | R | mAP50 |
|---|---|---|---|
| Diffusion+YOLOv10s | 60.0 | 44.3 | 48.5 |
| without PatchLoss | 62.6 | 47.3 | 51.6 |
| with PatchLoss | 64.0 | 48.1 | 52.7 |

**Comparison among different attention mechanisms.** The study compared the performance of the model with different attention mechanisms, such as Channel Attention (CA), Squeeze-and-Excitation (SE), and an Enhanced Multi-Attention (EMA) module. As illustrated in Table 6, the incorporation of this learning method yielded a noticeable improvement in the model's performance on the hazed COCO2017 dataset. The results indicated that the EMA module provided the best enhancement in terms of precision, recall, and mAP50, suggesting that it was most effective in capturing dependency relationships between adjacent channel features and improving the network's attention to targets.

**Influence of EMA attention position.** The EMA Attention module is a pivotal component of our model, specifically designed to augment the network's capability to capture multi-scale features. Its placement within the network architecture holds considerable sway over the model's overall performance.

To assess this, we conducted a series of experiments by inserting the EMA module at various positions within the YOLOv9s architecture. Specifically, we positioned the EMA module at the terminus of each Darknet block, at the junctions where blocks transition, and following the final convolutional layer. This approach enabled us to evaluate how the integration point of the EMA module impacts detection accuracy.

As shown in Table 7, the position of the EMA module is a determining factor in the model's performance. Our study found that by placing the EMA module in three different positions, Position 1: only in the Backbone section of Fig 3; Position 2: only in the Neck section of Fig 3; and Position 3: in both the Backbone and Neck sections of Fig 3, Position 3 yields the most favorable results, with improvements in P, R, and mAP50. The EMA module exerts its greatest efficacy when deployed after the network has extracted high-level features, enabling it to refine the feature representation before the detection head processes the information.

**Influence of wise-IoU.** The Wise-IoU loss function is a novel component introduced to address the issue of low-quality samples in the training data. It employs a dynamic non-monotonic focusing mechanism to improve the model's ability to learn from both high-quality and low-quality samples. To evaluate the effectiveness of the Wise-IoU loss, we compare the model's performance with and without this loss function.

**Table 6. Comparison among different attention mechanisms.**

| Method | P | R | mAP50 |
|---|---|---|---|
| YOLOv9s | 59.6 | 45.0 | 48.6 |
| YOLOv9s+CA | 60.1 | 46.2 | 49.8 |
| YOLOv9s+SE | 60.0 | 45.9 | 49.7 |
| YOLOv9s+EMA | 61.0 | 46.9 | 50.8 |

**Table 7. Influence of EMA attention.**

| Method | P | R | mAP50 |
|---|---|---|---|
| YOLOv9s | 59.6 | 45.0 | 48.6 |
| YOLOv9s+EMA(Position 1) | 60.3 | 46.2 | 49.7 |
| YOLOv9s+EMA(Position 2) | 60.7 | 46.6 | 50.2 |
| YOLOv9s+EMA(Position 3) | 61.0 | 46.9 | 50.8 |

As depicted in Table 8, the incorporation of the Wise-IoU loss significantly improves the model's performance, particularly in terms of recall and mAP50. This indicates that the Wise-IoU loss helps the model focus more effectively on the regions of interest, even when the training data contains low-quality samples.

Additionally, in the ablation study, we evaluate the robustness of the model by configuring different parameter settings for Wise-IoU. Five sets of parameter targets are tested, with alpha and beta set to 2.2 and 3, 1.6 and 3, 1.9 and 2, 1.9 and 3, and 1.9 and 4, respectively. Both horizontal and vertical comparisons are conducted. As shown in Fig 7, the configuration with alpha = 1.9 and beta = 3 achieves the best performance.

## Conclusion

This study introduces an innovative object detection framework designed to enhance visibility and accuracy in low-visibility haze conditions. The approach begins by applying haze to the COCO2017 dataset and then employs a dehazing module that leverages an improved GAN with an integrated patchwise contrastive loss function to generate dehazed images. These images are subsequently fed into an advanced YOLOv9s object detection model, which has been refined with the addition of the Efficient Multi-Scale Attention module and the WIoU loss function to boost detection precision. Comprehensive experiments conducted on the hazed COCO2017 dataset demonstrate the superiority of the proposed model in terms of detection accuracy. The results indicate that the model achieves state-of-the-art performance when compared to various existing techniques. Ablation experiments and comparative studies further confirm the effectiveness of each component within the proposed method.

The methodology encompasses multiple stages—namely, image dehazing and object detection. This sequential approach carries the risk of error propagation. Specifically, any inaccuracies or artifacts introduced during the image dehazing stage can inadvertently amplify in the subsequent object detection phase, ultimately detracting from the overall system's performance. This underscores the critical need for robust and precise image dehazing techniques that minimize error propagation.

The future directions of this paper worth further exploration are: (1) Further Enhancements in Image Dehazing: Although GAN-based image dehazing techniques have made significant progress, there is still room for improvement in preserving fine details and avoiding color distortions. Future work can explore advanced GAN architectures, loss functions, and regularization techniques to produce higher-quality dehazed images, further enhancing the

**Table 8. Influence of Wise-IoU.**

| Method | P | R | mAP50 |
|---|---|---|---|
| YOLOv9s | 59.6 | 45.0 | 48.6 |
| YOLOv9s+Wise-IoU | 61.5 | 46.5 | 50.6 |
| YOLOv9s+Wise-IoU+EMA | 62.6 | 47.3 | 51.6 |

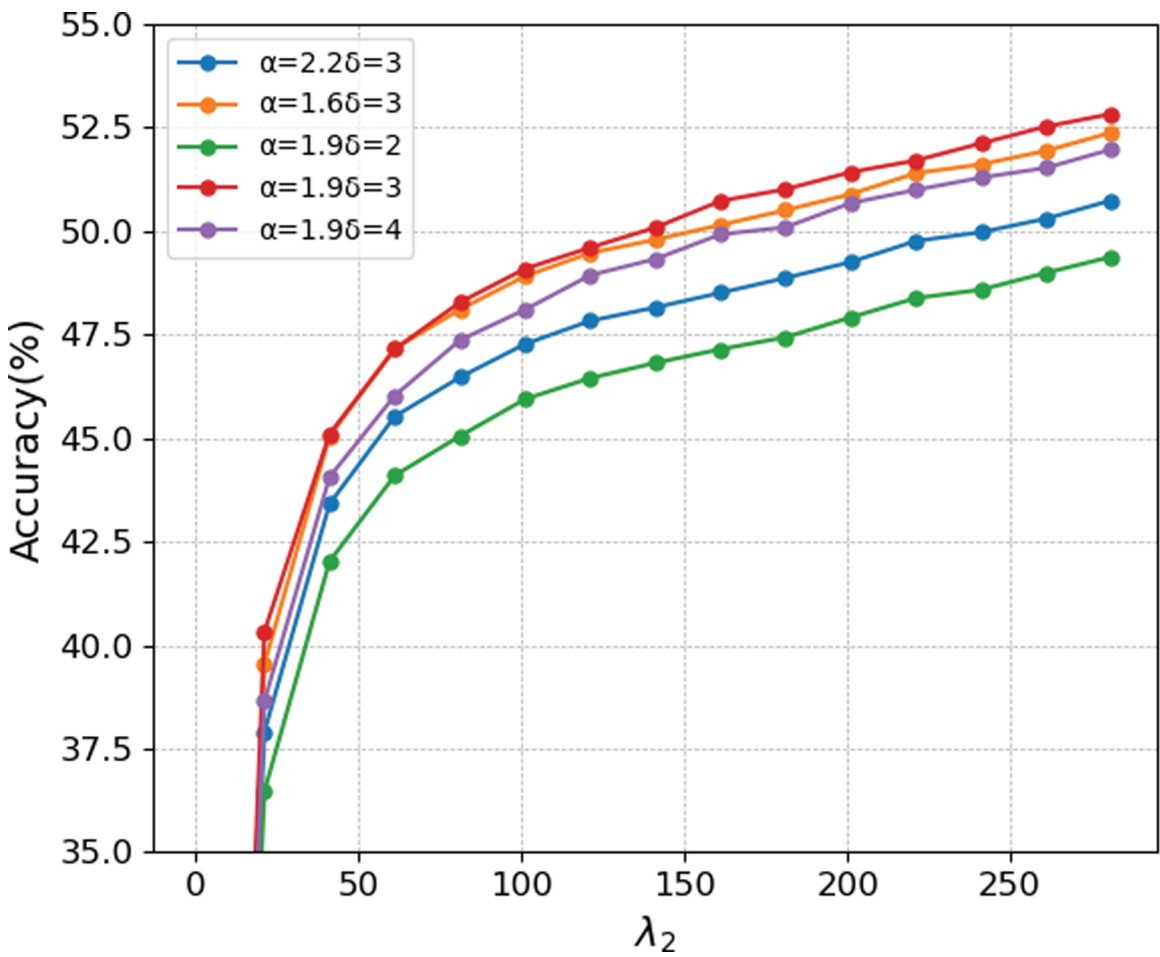

**Fig 7.** Training results under different parameters.

performance of downstream object detection tasks. (2) Optimization of Attention Mechanisms: The proposed EMA attention module has shown promising results in capturing multi-scale features. However, the optimal design and placement of attention mechanisms can vary based on the specific task and dataset. Future research can delve deeper into the optimization of attention mechanisms, including exploring combinations of different attention types and positions within the network architecture. Additionally, research attention mechanisms to enhance the focus on small and edge objects, aiming to improve overall prediction accuracy. (3) Unseen Data: To ensure the model's generalization ability, we plan to test the model in more real-world application scenarios in future work, providing a more comprehensive evaluation of its performance on unseen data.

## Author contributions

**Conceptualization:** Yang Zhang, Bin Zhou, Xue Zhao, Xiaomeng Song.

**Data curation:** Yang Zhang, Bin Zhou.

**Formal analysis:** Bin Zhou, Xue Zhao, Xiaomeng Song.

**Funding acquisition:** Bin Zhou.

**Investigation:** Yang Zhang, Bin Zhou, Xue Zhao, Xiaomeng Song.

**Methodology:** Yang Zhang, Bin Zhou.

**Project administration:** Yang Zhang, Bin Zhou.

**Resources:** Yang Zhang, Bin Zhou.

**Software:** Yang Zhang, Bin Zhou.

**Supervision:** Yang Zhang, Bin Zhou.

**Validation:** Xue Zhao, Xiaomeng Song.

**Visualization:** Xue Zhao, Xiaomeng Song.

**Writing – original draft:** Yang Zhang, Bin Zhou.

**Writing – review & editing:** Yang Zhang, Bin Zhou.

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
