## [Decision Letter · Decision Letter 0]

21 Aug 2024

PONE-D-24-31889

Enhanced Object Detection in Low-Visibility Haze Conditions with YOLOv9s PLOS ONE

Dear Dr. Zhou,

Thank you for submitting your manuscript to PLOS ONE. After careful consideration, we feel that it has merit but does not fully meet PLOS ONE’s publication criteria as it currently stands. Therefore, we invite you to submit a revised version of the manuscript that addresses the points raised during the review process.

We look forward to receiving your revised manuscript.

Kind regards,

Xiongkuo Min

Academic Editor

PLOS ONE

Journal Requirements:

"the Shandong Provincial Natural Science Foundation of China"

5. We note that Figures 1, 6 and 7 in your submission contain copyrighted images. All PLOS content is published under the Creative Commons Attribution License (CC BY 4.0), which means that the manuscript, images, and Supporting Information files will be freely available online, and any third party is permitted to access, download, copy, distribute, and use these materials in any way, even commercially, with proper attribution. For more information, see our copyright guidelines: http://journals.plos.org/plosone/s/licenses-and-copyright.

a. You may seek permission from the original copyright holder of Figures 1, 6 and 7 to publish the content specifically under the CC BY 4.0 license. 

Reviewers' comments:

Reviewer's Responses to Questions

**Comments to the Author**

1. Is the manuscript technically sound, and do the data support the conclusions?

Reviewer #1: Yes

Reviewer #2: Partly

Reviewer #3: Yes

Reviewer #4: Partly

2. Has the statistical analysis been performed appropriately and rigorously? 

Reviewer #1: Yes

Reviewer #2: No

Reviewer #3: Yes

Reviewer #4: No

3. Have the authors made all data underlying the findings in their manuscript fully available?

Reviewer #1: Yes

Reviewer #2: Yes

Reviewer #3: Yes

Reviewer #4: Yes

4. Is the manuscript presented in an intelligible fashion and written in standard English?

Reviewer #1: Yes

Reviewer #2: Yes

Reviewer #3: Yes

Reviewer #4: Yes

5. Review Comments to the Author

Reviewer #1: In this paper, authors introduce an innovative object detection framework designed to enhance

visibility and accuracy in low-visibility haze conditions. The approach begins by applying haze to the COCO dataset and then employs a dehazing module that leverages an improved GAN with an integrated patchwise contrastive loss function to generate dehazed images. These images are subsequently fed into an advanced YOLOv9s object detection model, which has been refined with the addition of the Efficient Multi-Scale Attention module and the Wise-IoU loss function to boost detection precision. Comprehensive experiments conducted on the hazed COCO dataset demonstrate the superiority of the proposed model in terms of detection accuracy. However, there are some major issues, which need to be handled in the text properly:

Authors compare the proposed technique with existing techniques in terms of detection accuracy. Authors should also compare the proposed technique with existing techniques in terms of computation times, training times and storage sizes.

The failure case of the proposed technique must be described and discussed in the text.

In this paper, authors propose a technique to alleviate the nonuniform illumination obtained with non-collimated light sources performing Reflectance Transformation Imaging (RTI) acquisitions. It’s shown that it is possible to obtain a per-pixel estimation of light-surface distance as well as light angular position knowing the real size of the surface as well as the light distance and angle at the center of the surface. It’s also shown that, using the estimated distances and the elevation angles, it is possible to obtain adjustment coefficients following the illumination model of a point light source. Authors show the efficiency of the proposed method on RTI acquisitions performed on cultural heritage objects and a manufactured surface. Authors show that the proposed method corrects the effects of non-uniform illumination and leads to improve the relighting commonly associated with RTI. However, there are some major issues, which need to be handled in the text properly:

Authors should compare the proposed technique with existing techniques both numerically and visually.

The failure case of the proposed technique must be described and discussed in the text.

The source code of the proposed technique must be shared to implement it easily.

Authors should cite the following recent object detection paper [Azadvatan2024arXiv].

At the end of Conclusion section, authors can also give some important future directions to the readers, researchers.

@article{Azadvatan2024arXiv,

author = {{Azadvatan}, Yashar and {Kurt}, Murat},

title = {MelNet: A Real-Time Deep Learning Algorithm for Object Detection},

journal = {arXiv preprint arXiv:2401.17972},

pages = {arXiv:2401.17972},

year = {2024},

month = jan,

eid = {arXiv:2401.17972},

doi = {10.48550/arXiv.2401.17972},

url = {https://doi.org/10.48550/arXiv.2401.17972},

archivePrefix = {arXiv},

eprint = {2401.17972},

primaryClass = {cs.CV},

keywords = {Computer Science - Computer Vision and Pattern Recognition, Computer Science - Artificial Intelligence, Computer Science - Machine Learning}}

Reviewer #2: The paper titled "Enhanced Object Detection in Low-Visibility Haze Conditions with YOLOv9s" presents a novel framework for object detection in challenging haze environments. The proposed model improves detection accuracy by integrating contrastive learning, multi-scale attention mechanisms, and dynamic focusing techniques, particularly enhancing the YOLOv9s architecture.

However, there are several issues that need to be addressed:

1-There is no comparison between the proposed method and other state-of-the-art methods specifically designed for haze conditions. The authors should include a more detailed discussion and comparative analysis, explaining why the proposed method outperforms others.

2-I suggest covering more benchmark metrics for comparison.

3-The conclusion of this paper lacks future directions or suggestions, which are necessary for guiding further research.

4-The manuscript could be strengthened by including experiments on additional datasets, particularly those that capture real-world haze conditions. Alternatively, the authors could discuss the potential limitations of the current approach when applied to real-world data and suggest future research directions to address these challenges.

Reviewer #3: This paper presents a new method for object detection in hazy scenes.

The proposed method seems reasonable, and the paper is well written, but the experimental validation needs some improvement.

In the introduction and related work sections, some of the relevant works are missing, especially the works about the evaluation of haze and dehazing, for example, HazDesNet: an end-to-end network for haze density prediction; Quality evaluation of image dehazing methods using synthetic hazy images; Objective quality evaluation of dehazed images.

“…This degradation in image quality can severely impact the accuracy of computer vision tasks, such as object detection…”

“…Eliminating haze from images not only improves image quality but also enhances information readability…”

Some related surveys are suggested to be given for better referring of the related topics, for example visual quality as its evaluation as discussed in the related surveys, e.g., ‘Perceptual image quality assessment: a survey’, ‘Perceptual video quality assessment: a survey’, and ‘Screen content quality assessment: Overview, benchmark, and beyond’.

The experimental validation needs to be strengthened. For example, the combinations of SOTA dehazing methods and SOTA detection methods are also suggested to be included into the comparison.

The formats of some references are suggested to be double-checked.

Reviewer #4: Strengths:

- The manuscript is clearly written, with a logical structure that aids in understanding.

- Complex ideas are conveyed in a manner that is accessible to a wide audience.

Weaknesses:

- The experimental design is not detailed enough to substantiate the theoretical claims, with inadequate information in the code and experiment sections.

- The paper lacks a focus on parameter fine-tuning, which is critical for evaluating the model's robustness.

- There is no discussion of the computational costs, especially during inference, which is an important aspect to consider.

- The literature review is not comprehensive enough, failing to provide sufficient comparisons with established methods.

- The absence of standard deviations in the reported results undermines their statistical validity.

- The manuscript does not address how the model generalizes to unseen data, which is crucial for its practical use.

- There is a lack of discussion on potential limitations and areas for future research, leaving the reader with an incomplete picture of the work's scope.

Additional Comment: What exactly sets your novel method apart? Is it akin to a hypothetical 'YOLOv9'? The paper claims to enhance object detection in low-visibility haze conditions with YOLOv9s, but it is essential to validate this claim by using more public datasets. Proving that the proposed model is truly the best requires thorough comparative experiments.

The most significant revisions will likely revolve around addressing the concerns outlined above. In general, I anticipate:

- A more thorough discussion and comparison with existing work;

- Additional experiments with detailed statistical analysis;

- Refinements to the discussion to better evaluate the strengths and limitations of the approach.

Summary of Reasons for Rejection:

Significant revisions are required, including a more thorough experimental design, detailed statistical reporting, and discussions on generalization, limitations, and future work. These improvements are necessary for the manuscript to be reconsidered.

6. PLOS authors have the option to publish the peer review history of their article (what does this mean?). If published, this will include your full peer review and any attached files.

Reviewer #1: No

Reviewer #2: No

Reviewer #3: No

Reviewer #4: **Yes: **Teerapong Panboonyuen

---

## [Author Response · Author response to Decision Letter 1]

17 Nov 2024

Dear Editor,

Thank you for allowing a resubmission of our manuscript, with an opportunity to address the reviewers’ comments. In particular, in order to avoid copyright problems of the original images, all original images have been deleted from the manuscript we have submitted this time, and the drawings in the manuscript are the schematic diagram of the method proposed in the paper. Moreover, we have given a detailed reply to our revision in the attachment "response to reviewers". We sincerely hope that the journal will see that we put our best effort into every comment of the journal and the reviewers.

We are uploading (a) our point-by-point response to the comments (Response to Reviewers) and (b) an updated manuscript with blue highlighting indicating changes (Revised Manuscript with Track Changes) and (c) an updated manuscript without blue highlighting indicating changes (Manuscript).

---

## [Decision Letter · Decision Letter 1]

3 Dec 2024

PONE-D-24-31889R1
Enhanced Object Detection in Low-Visibility Haze Conditions with YOLOv9s
PLOS ONE

Dear Dr. Zhou,

Thank you for submitting your manuscript to PLOS ONE. After careful consideration, we feel that it has merit but does not fully meet PLOS ONE’s publication criteria as it currently stands. Therefore, we invite you to submit a revised version of the manuscript that addresses the points raised during the review process.

We look forward to receiving your revised manuscript.

Kind regards,

Xiongkuo Min

Academic Editor

PLOS ONE

Journal Requirements:

Reviewers' comments:

Reviewer's Responses to Questions

**Comments to the Author**

1. If the authors have adequately addressed your comments raised in a previous round of review and you feel that this manuscript is now acceptable for publication, you may indicate that here to bypass the “Comments to the Author” section, enter your conflict of interest statement in the “Confidential to Editor” section, and submit your "Accept" recommendation.

Reviewer #1: All comments have been addressed

Reviewer #3: All comments have been addressed

2. Is the manuscript technically sound, and do the data support the conclusions?

Reviewer #1: Yes

Reviewer #3: Yes

3. Has the statistical analysis been performed appropriately and rigorously? 

Reviewer #1: No

Reviewer #3: Yes

4. Have the authors made all data underlying the findings in their manuscript fully available?

Reviewer #1: Yes

Reviewer #3: Yes

5. Is the manuscript presented in an intelligible fashion and written in standard English?

Reviewer #1: Yes

Reviewer #3: Yes

6. Review Comments to the Author

Reviewer #1: In this paper, authors introduce an innovative object detection framework designed to enhance visibility and accuracy in low-visibility haze conditions. The approach begins by applying haze to the COCO dataset and then employs a dehazing module that leverages an improved GAN with an integrated patchwise contrastive loss function to generate dehazed images. These images are subsequently fed into an advanced YOLOv9s object detection model, which has been refined with the addition of the Efficient Multi-Scale Attention module and the Wise-IoU loss function to boost detection precision. Comprehensive experiments conducted on the hazed COCO dataset demonstrate the superiority of the proposed model in terms of detection accuracy. In the revised paper, it seems that authors handled most of the reviewer comments into account. However, there are some minor issues, which need to be handled in the text properly:

Authors compare the proposed technique with existing techniques numerically. Authors should also compare the proposed technique with existing techniques visually.

Reviewer #3: Most of the reviewer's previous concerns are addressed. The paper is now suggested to be published.

7. PLOS authors have the option to publish the peer review history of their article (what does this mean?). If published, this will include your full peer review and any attached files.

Reviewer #1: No

Reviewer #3: No

---

## [Author Response · Author response to Decision Letter 2]

3 Jan 2025

Dear editor,

We have included visualizations in our "response to reviewers" that compare our method with others on the COCO dataset test set. These visualizations intuitively showcase the detection results of our method versus others. We randomly select five test images from the COCO dataset to create five sets of visualization results, which are displayed in "response to reviewers" in detail. For example, in the first set, our method achieves better accuracy compared to other methods and does not experience missed detections as seen in methods such as YOLOv8s and YOLOv5s. Additionally, compared to the original labels, our method does not make any incorrect predictions. From these visualizations, it is evident that our method achieves better overall detection accuracy and has lower probabilities of missed and false detections compared to other methods. However, we don’t include these images in our paper; instead, they are available on our open-source GitHub repository at the link (https://github.com/PaTinLei/EOD) for readers’ reference and study.

Next, we will explain why we don’t include these visualizations in the paper.

Firstly, in the previous revision responses, we were required to provide CC BY 4.0 license declarations for all copyrighted images. We made various efforts, including but not limited to researching materials, contacting dataset officials, providing CC BY 2.0 licenses, and replacing datasets. However, we were ultimately unable to obtain the CC BY 4.0 license for the datasets involved in our method. This is difficult because these public datasets are released without declaring a CC BY 4.0 license, and their team only offers a CC BY 2.0 license. To facilitate our paper's swift entry into the review process, we have to remove all images involving copyright issues from the manuscript.

Secondly, we also considered creating our own dataset. However, producing a dataset ourselves would require a significant amount of time for data collection and annotation, which is overly burdensome for an individual. Furthermore, our own dataset would not be as reference-worthy or reliable as public datasets. Moreover, we believe that even our own dataset would still present copyright issues, such as copyrighted elements within the images.

Finally, considering the reasons mentioned above, we don’t include images with copyright disputes in our paper. We hope that by providing this response, we can ensure a clear presentation of the visual results from our manuscript.

With best regards!

Yours sincerely

Zhou,

Corresponding author: Bin Zhou

E-mail: freetzb@163.com

---

## [Editor Report · Decision Letter 2]

7 Jan 2025

Enhanced Object Detection in Low-Visibility Haze Conditions with YOLOv9s

PONE-D-24-31889R2

Dear Dr. Zhou,

We’re pleased to inform you that your manuscript has been judged scientifically suitable for publication and will be formally accepted for publication once it meets all outstanding technical requirements.

Kind regards,

Xiongkuo Min

Academic Editor

PLOS ONE
---

## [Editor Report · Acceptance letter]

PONE-D-24-31889R2

PLOS ONE

Dear Dr. Zhou,

I'm pleased to inform you that your manuscript has been deemed suitable for publication in PLOS ONE. Congratulations! Your manuscript is now being handed over to our production team.

Kind regards,

on behalf of

Dr. Xiongkuo Min

Academic Editor

PLOS ONE